# A Deep Learning Approach for Predicting Multiple Sclerosis

**DOI:** 10.3390/mi14040749

**Published:** 2023-03-29

**Authors:** Edgar Rafael Ponce de Leon-Sanchez, Omar Arturo Dominguez-Ramirez, Ana Marcela Herrera-Navarro, Juvenal Rodriguez-Resendiz, Carlos Paredes-Orta, Jorge Domingo Mendiola-Santibañez

**Affiliations:** 1Facultad de Informática, Universidad Autónoma de Querétaro, Querétaro 76230, Mexico; 2Centro de Investigación en Tecnologías de Información y Sistemas, Universidad Autónoma del Estado de Hidalgo, Pachuca 42039, Mexico; 3Facultad de Ingeniería, Universidad Autónoma de Querétaro, Querétaro 76010, Mexico; 4Centro de Investigaciones en Óptica, Aguascalientes 20200, Mexico

**Keywords:** deep learning, artificial neural network, multiple sclerosis

## Abstract

This paper proposes a deep learning model based on an artificial neural network with a single hidden layer for predicting the diagnosis of multiple sclerosis. The hidden layer includes a regularization term that prevents overfitting and reduces the model complexity. The purposed learning model achieved higher prediction accuracy and lower loss than four conventional machine learning techniques. A dimensionality reduction method was used to select the most relevant features from 74 gene expression profiles for training the learning models. The analysis of variance test was performed to identify the statistical difference between the mean of the proposed model and the compared classifiers. The experimental results show the effectiveness of the proposed artificial neural network.

## 1. Introduction

Multiple sclerosis (MS) is a chronic inflammatory disease of the central nervous system (CNS) of autoimmune etiology, characterized by localized areas of demyelination, axonal loss, and gliosis in the brain and spinal cord [1]. MS can be classified into three types based on its progression: primary progressive MS (PPMS), relapsing-remitting MS (RRMS), and secondary progressive MS (SPMS) [2]. The most common type is RRMS, accounting for 80% of MS patients. Susceptibility to MS is complex but involves environmental events, and genetic factors [3]. On the genetic side, several genome-wide association screens (GWAS), which incorporate large arrays of single nucleotide polymorphisms (SNPs), have now identified many common MS-risk variants located in scattered genomic regions as being associated with MS [4]. Although MS has a complex etiology, human leukocyte antigen (HLA) genes have been implicated in disease susceptibility for four decades. HLA-Class II alleles represent the more significant genetic contribution to MS risk, specifically within the DR15 haplotype: HLA-DRB1*15:01, is a common finding in MS populations, primarily those of Northern European descent [5].

In the last decade, there has been a significant increase in machine learning (ML) applications studying neurological diseases. ML algorithms are data science approaches to build predictive models that can learn patterns and relationships within data while requiring minimal human intervention [6]. The ML application in MS thus far has mainly been for classifying participants into the different disease stages (clinically isolated syndrome (CIS), RRMS, SPMS, among others), for predicting the diagnosis of MS, for predicting the transition from CIS to clinically definite MS, for predicting disability progression, and for predicting the patient’s possible response to pharmacological therapy to help the professional in choosing the most appropriate treatment [7]. However, there is no single clinical study or laboratory finding that can secure a definitive diagnosis of MS. The diagnosis is made based on consensus clinical, imaging, and laboratory criteria [8]. Some studies have focused on the diagnosis of MS using different blood serum markers [9]. Goyal et al. [10] analyzed the serum level of eight cytokines: IL-1β, IL-2, IL-4, IL-8, IL-10, IL-13, IFN-γ, and TNF-α in MS patients, to identify predictors of disease. The datasets were used as input into four learning models. Random forest (RF) was identified as the best model for MS diagnosis as it performed remarkably on all the considered criteria. In this paper, a deep learning (DL) model based on an artificial neural network (ANN) with a single hidden layer is proposed for predicting the diagnosis of MS in 144 individuals, 99 MS, and 45 healthy controls, using their mRNA expression profiling as predictors. An extra model is conformed adding a second hidden layer to the network structure, in order to analyze whether a network with two hidden layers and fewer hidden neurons achieves higher performance and a lower error rate. A comparison of the prediction performance of the proposed ANN model and four conventional ML techniques was performed. Casalino et al. [11] published a classification study to evaluate the effectiveness of three ML methods in distinguishing pediatric MS from healthy children based on their miRNA expression profiling. Encouraging results were obtained with a multi-layer perceptron (MLP) model based on a set of features selected by a support vector machine (SVM) algorithm. Chen et al. [12] integrated three peripheral blood mononuclear cell (PBMC) microarray datasets and one peripheral blood T-cells microarray dataset, which allowed a comprehensive analysis of the biological functions of MS-related genes. Differential expression analysis identified 78 significantly expressed genes in MS. A subsequent analysis identified the CXCR4, ITGAM, ACTB, RHOA, RPS27A, UBA52, and RPL8 genes as potential biomarkers associated with MS diagnosis. An SVM was employed to establish an MS diagnostic model with high prediction performance on different dataset platform chips. Among the studies that suggest that genetics can predict the possible patient response to treatment, Fagone et al. [13] applied the uncorrelated reduced centroid algorithm (UCRC), to identify a subset of genes that could predict the pharmacological response to natalizumab treatment between RRMS patients. The results suggest that a specific gene expression profiling of CD4+T cells can characterize the responsiveness to natalizumab. Jin et al. [14] proposed a bio-informatic feature selection procedure to identify gene pairs with differentially correlated edges (DCE). The proposed method was applied to a microarray data set to evaluate the effect of IFN-β treatment in RRMS patients. Among 23 identified genes, seven had a confidence score >2: CXCL9, IL2RA, CXCR3, AKT1, CSF2, IL2RB, and GCA. An SVM model was trained with these genes and had good predictive results. While the data volume is complex and multi-dimensional, there is much redundancy and irrelevant information. Feature selection is a fundamental data dimensionality reduction technique often used in ML and DL [15]. Selecting the features can significantly improve the computational efficiency of the classification or regression algorithms while increasing the learning model’s performance. In this paper, a feature selection method based on recursive feature elimination with cross-validation [16] is performed to find the optimal number of relevant features in 74 gene expression profiles related to MS. Algorithms based on metaheuristic methods have demonstrated an ability to search for suitable subsets of features for optimization problems. For feature selection, Aviles et al. [17] proposed a methodology based on genetic algorithms to find the parameter space that offers the slightest classification error to improve the electromyography (EMG) process.

The complexity of a problem implicitly refers to the complexity of an algorithm for solving that problem, and to the measure of complexity that allows to evaluate the algorithm’s performance [18]. Two different kinds of complexity measures can be identified: statics based only on the structure of the algorithms, and dynamics that considers both the algorithms and the inputs, and are thus based on the behavior of a computation. Achache [19] dealt with the study of the polynomial complexity and numerical implementation for a short-step primal-dual interior point algorithm for monotone linear complementarity problems (LCP). In this paper, an algorithms complexity analysis based on two typical static measures: runtime, and program size, is performed. Additionally, the statistical hypothesis test (ANOVA) is computed to analyze the statistical difference between the mean of the proposed ANN model and the compared classifiers. Salamai et al. [20] implemented this statistical test to identify the operational risks in the supply chain 4.0 based on a Sine Cosine Dynamic Group (SCDG) algorithm, obtaining satisfactory results.

This paper is organized as follows. Section 2 explains the proposed research strategy. Section 3 provides the experimental results. Section 4 discusses the proposed ANN model. Finally, Section 5 presents the conclusions of the study.

## 2. Materials and Methods

A flowchart about the strategy followed in this research is shown in Figure 1, which divides the proposal into five stages.

### 2.1. Data Import

The dataset was collected from the GSE17048 expression profiling by array experiment, available in the public repository of genomic data GEO [21]. Through the GPL6947 platform (Illumina HumanHT-12 V3.0 expression beadchip), the mRNA expression profiling of 74 genes were acquired from 144 individuals, 99 with MS (43 PPMS, 36 RRMS, and 20 PSMS) and 45 healthy controls. The complete dataset is composed of the HLA-DRB1 gene, because it has a deep link to the risk of MS [5], and 73 expression profiles were taken as a reference from the 78 MS-related genes identified by Chen et al. [12], of which five were not considered. The expression summary values were analyzed by GEO2R, an interactive web tool that allows viewing a specific gene expression through the profile graph tab. The expression values of the genes across the samples are displayed and presented as a table of genes ordered by significance, and then they are integrated into an excel spreadsheet.

### 2.2. Data Preprocessing

Standardization: this technique normalizes the features by removing the mean and scaling to unit variance [11]. Overfitting is a common problem in ML and DL, where a model works well on the training data but not on the testing data, i.e., the model is too complex with a high variance [22]. To avoid overfitting, the input data are divided into 80% training (X_train) and 20% testing (X_test), based on Pareto analysis [23]. Additionally, the output labels are separated into 80% y_train and 20% y_test for validation. After dividing the dataset, X_train and y_train are standardized.Feature selection: in linear models, the target value is modeled as a linear combination of the features [24]. After standardizing the training data, the dimensionality reduction technique: recursive feature elimination (RFE) with cross-validation is used to select the most important features [16]. Given an external estimator that assigns weights to features (for example, the coefficients of a linear model), the RFE goal is to select features recursively, considering smaller and smaller sets of them. First, the estimator is trained on the initial set of features. The importance of each one is obtained either through any specific attribute, such as coefficients value (weights assigned to the features, coef_) or feature importances (the impurity-based feature importances, feature_importances_). Then, the least important features are pruned from the current set. That procedure is recursively repeated on the pruned set until the desired number of elements to select is eventually reached. RFE with cross-validation (RFECV) performs RFE in a cross-validation loop to find the optimal number of features. The scoring strategy ”accuracy” optimizes the proportion of correctly classified samples.

### 2.3. Training and Classification

Machine learning models: The K-Neighbors (KN) [25], Gaussian Naive Bayes (GNB) [26], C-Support Vector (CSV) [27], and Decision Tree (DT) [28] techniques are trained with the most relevant genetic features selected by RFECV method. The Anaconda 3 2021.05 (Python 3.8.8 64-bit) software and the open-source development internet application Jupyter Notebook are used to generate the pseudo codes that are executed on a personal computer with Windows 10 Home, 11th Gen Intel Core i5-1135G7 2.4 GHz processor, 8 GB memory, and 500 GB hard disk. Hyperparameters are the settings that can be arbitrarily configured before starting the training process to optimize the model performance, e.g., in Random Forest-based algorithms, the number of estimators (number of decision trees) and the criterion or impurity measure. In contrast, model parameters, such as weights in neural networks, are learned during the model training process [29]. The hyperparameters of the four ML techniques are set by default.Deep learning models: at the core of DP are neural networks, mathematical entities capable of representing complex functions through a composition of simple functions. The basic building block of these complex functions is the neuron. It is a linear transformation of the input (for example, multiplying the input by a number, the weight, and adding a constant, the bias) followed by applying a fixed nonlinear function, the activation function [30]. Mathematically, the neuron output can be expressed as o=f(w∗x+b), with *x* as the input, *w* as the weight or scaling factor, and *b* as the bias or offset. *f* is the activation function, commonly set to hyperbolic tangent. A multi-layer neural network is a composition of functions such as Equations (Equation 1)–(Equation 4).
(1)$$x_{1}=f\left(w_{0}*x+b_{0}\right)$$
(2)$$x_{2}=f\left(w_{1}*x_{1}+b_{1}\right)$$
(3)…
(4)$$y=f(w_{n}*x_{n}+b_{n})$$The output of a layer of neurons is used as an input for the following layer. Between the input, and the output layer, there can be one or more non-linear layer, called hidden layers. The leftmost layer or input layer, consists of a set of neurons representing the input features. The output layer receives the values from the last hidden layer and transforms them into output values.The number of hidden neurons $N_{h}$ can be determined by Equation (Equation 5),
(5)$$N_{h}=\frac{N_{in}+\sqrt{N_{p}}}{L}$$
where $N_{in}$ is the number of input neurons, $N_{p}$ the number of input samples, and *L* the number of hidden layers [31].The proposed ANN architecture is presented in Figure 2, where 144 is the number of individuals, 35 is the number of input neurons (features selected by RFECV method), 106 is the number of computed hidden neurons of a single dense-type hidden layer with ’tanh’ as the activation function, followed by a dropout-type layer with 0.1 frequency. The second dense layer with ’sigmoid’ as activation function receives the values from the dropout layer and transforms them into output predictions (healthy/MS). The number of hidden layers is set to one for comparison purposes. An extra model is conformed by adding a second hidden layer to the network structure, in order to analyze whether a network with two hidden layers and fewer hidden neurons (53 units) than a single hidden layer (106 units) achieves higher performance and lower validation loss [32].In addition, the dense layer includes a kernel regularizer argument (kernel_regularizer = l2 with learning rate, lr = 0.01), which implements a regularizer function applied to the kernel weights matrix. The l2 regularization prevents overfitting and reduces model complexity.

### 2.4. Performance Metrics

The confusion matrix (CM), accuracy, sensitivity, specificity, logistic loss (log loss) or cross-entropy loss, and area under the curve (AUC) metrics [10,20,22] are computed to measure the predictive performance of the compared classifiers.

### 2.5. Statistical Analysis

The analysis of variance or ANOVA test is applied to identify the statistical difference between the mean of the proposed ANN model and the compared classifiers [20]. Two hypotheses, the null and the alternative one, are formulated. The null hypothesis is $H_{0}:\mu \left(KN\right)=\mu \left(GNB\right)=\mu \left(CSV\right)=\mu \left(DT\right)=\mu \left(ANN^{1}\right)=\mu \left(ANN^{2}\right)$ where μ is the mean of samples, and the alternate hypothesis is $H_{1}$: non-equal means. The *p*-value is the significance level that shows whether there are significant differences between the means of the data.

## 3. Results

In this paper, a performance comparison of the proposed ANN model and four conventional ML techniques are carried out. The most relevant features from 74 genes related to MS etiology were used as training inputs for predicting the susceptibility to the disease.

### 3.1. Feature Selection

Figure 3 displays the feature importance results provided by an RF estimator.

Table 1 presents the selected features by RFECV method based on the highest importance score. The number of selected features was optimized using the accuracy scoring strategy. The model with 35 features is optimal, presenting the highest accuracy achieved, 1.0 training accuracy, and 0.75 test accuracy. After the selection, the remaining 39 features were excluded.

The learning models were trained with and without feature selection for analyzing the computational efficiency and the algorithm’s complexity. Table 2 shows the results of efficiency, based on a less runtime and less memory (dataset file), and complexity, based on a larger runtime and larger program size. So, feature selection increased the efficiency of all the compared classifiers. The complexity of ANN1 and ANN2 algorithms was superior to the four ML algorithms.

### 3.2. Performance Comparison

The KN, GNB, CSV, DT, and ANN learning models were trained with 35 selected features by the RFECV method. Then, the CM, accuracy, sensitivity, specificity, logistic loss, and AUC metrics were computed with the output predictions for comparing the classifiers performance.

The input data (5040 samples) were divided into 80% X_train (4032 samples) and 20% X_test (1008) to avoid overfitting. In addition, the output labels (144) were divided into 80% y_train (115) and 20% y_test (29) for validation. The CM results of the proposed ANN with a single hidden layer represent seven individuals predicted as negative (healthy), 19 individuals predicted as positive (MS) correctly, two individuals predicted as negative (healthy) incorrectly, and one individual predicted as positive (MS) wrongly.

The results of the remaining performance metrics are presented in Table 3. Feature selection improved the accuracy score of almost all classifiers.

A comparative graph of the performance results of Table 3 is shown in Figure 4.

In the case of the proposed ANN, Figure 5 displays the training and validation accuracy and loss results by several hidden layers. ANN with a single hidden layer achieved the highest validation accuracy and the least validation loss.

## 4. Discussion

ML and DL are based on mathematical algorithms that find natural patterns in the data, and they are emerging as very useful tools in the bio-informatics field [7]. These classification models can be trained with gene expression data to improve the diagnosis of some diseases, e.g., early MS [10,11,12], and help specialists to select the most appropriate therapy for a individual patient [13,14]. In this paper, a DL model based on an ANN with a single hidden layer was proposed for predicting the diagnosis of MS. As Table 3 shows, higher prediction accuracy and a minimum loss were achieved compared with the four conventional ML techniques. Therefore, the proposed ANN model can be an option in providing short-term predictions of the susceptibility to MS based on individual’s genetics. Moreover, it provides a new understanding of the etiology of MS and can be a valuable support to specialists. To choose the correct number of hidden layers, for this particular case of research, it was proven that a network with a single hidden layer is better than with two hidden layers, because a network with a single hidden layer and more hidden neurons achieved a higher validation accuracy, in addition, the validation loss parameter converges faster, as Figure 5 shows.

The human genome is complex, and its volume is multi-dimensional. So, it requires the application of a dimensionality reduction method that allows us to ignore irrelevant information, improve the computational efficiency and increase the performance of the learning models. Hence, the RFECV method was applied to select the 35 most relevant features from 74 genes related to MS [12]. This method was chosen because it finds the optimal number of features based on the highest accuracy achieved. From the results in Table 2 and Table 3, the feature selection improved the computational efficiency (runtime and memory) and the prediction accuracy of the compared learning models. Regarding the complexity (runtime and program size) of DL algorithms, it was larger than ML algorithms.

The ANOVA test was performed to analyze the statistical difference between the mean of the proposed ANN model and the compared classifiers. Table 4 displays the descriptive statistics of data.

Table 5 presents the ANOVA test results, which show that the differences between the means are statistically significant (p<0.05), hence, the alternative hypothesis $H_{1}$ was accepted.

The experimental results obtained in this research indicate the effectiveness of the proposed ANN model, which can be a reference for future comparisons, using another learning techniques and identifying training data from another genes related to MS.

## 5. Conclusions

Some ML applications in MS have been proposed by researchers for predicting disease diagnosis using different genetic biomarkers. In this research paper, an ANN model is trained with 35 relevant genetic features related to MS. A 0.8965 accuracy and a 3.573 log loss were achieved compared with four conventional learning techniques. Thus, the DL models significantly increase the prediction accuracy and diminish the prediction loss compared with ML models. Hence, the proposed ANN model has a high potential of clinical application to support specialists in predicting the diagnosis of MS based on individual’s genetic features, allowing the emergence of new preventive treatments. To reduce the computational cost, the relevant features from 74 genetic expression profiling were selected by the RFECV method with 1.0 training accuracy and 0.75 test accuracy. So, the 35 selected features of Table 1 can be convenient predictive biomarkers for improving the comprehension of the influence of some genes on the susceptibility to MS, and play a significant role in comprehending the MS etiology. The results obtained from the ANOVA test confirm that the differences between the mean of the proposed ANN model and the compared classifiers are statistically significant based on *p*-value score (*p* < 0.05).

## Figures and Tables

**Figure 1 micromachines-14-00749-f001:**
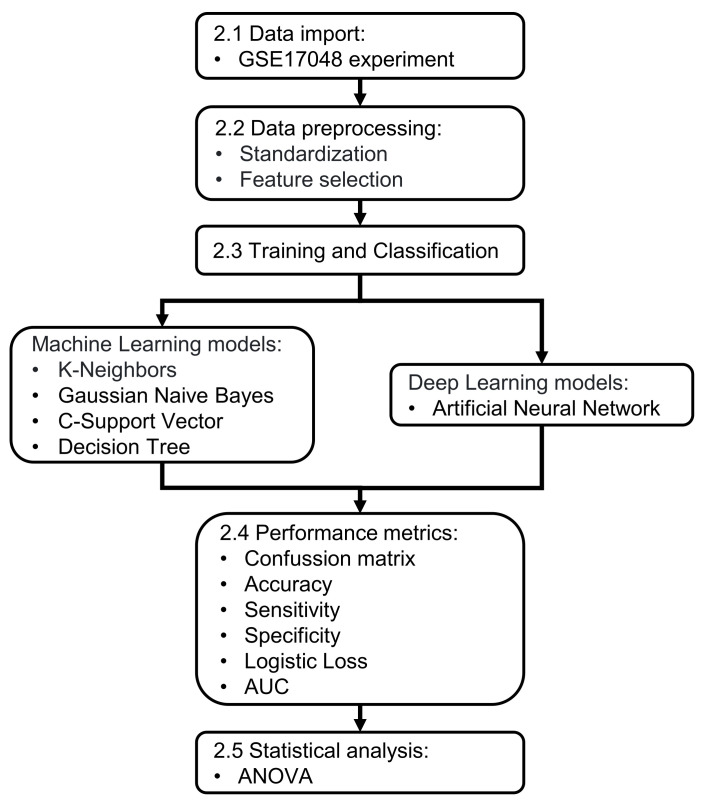
Proposed methodology. The gene data are obtained from the database and posteriorly standardized. The most relevant features are selected to train the compared prediction models and classify the individuals: healthy/MS. Then, the performance metrics are computed with the obtained predictions and the results are compared. Finally, the ANOVA test is performed to validate the effectiveness of the proposed ANN model.

**Figure 2 micromachines-14-00749-f002:**
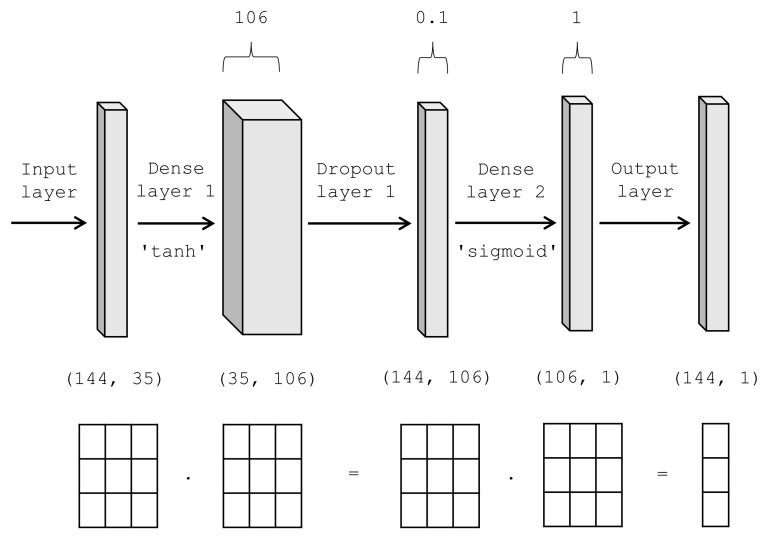
Proposed ANN architecture; dense layer implements the operation: output=activation(dot(input,kernel)+bias), where activation is the element-wise activation function, kernel is a weights matrix, and bias is a bias vector; dropout is a regularization layer that randomly sets input units to 0 with a frequency of rate at each step during training time, which helps prevent overfitting.

**Figure 3 micromachines-14-00749-f003:**
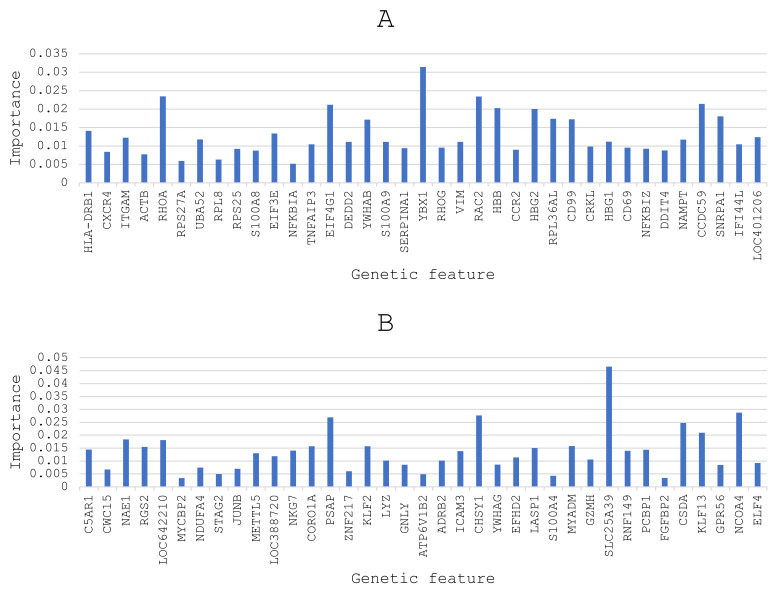
Feature importances; this graph presents the 74 genes divided into two blocks; (**A**): first part; (**B**): second part.

**Figure 4 micromachines-14-00749-f004:**
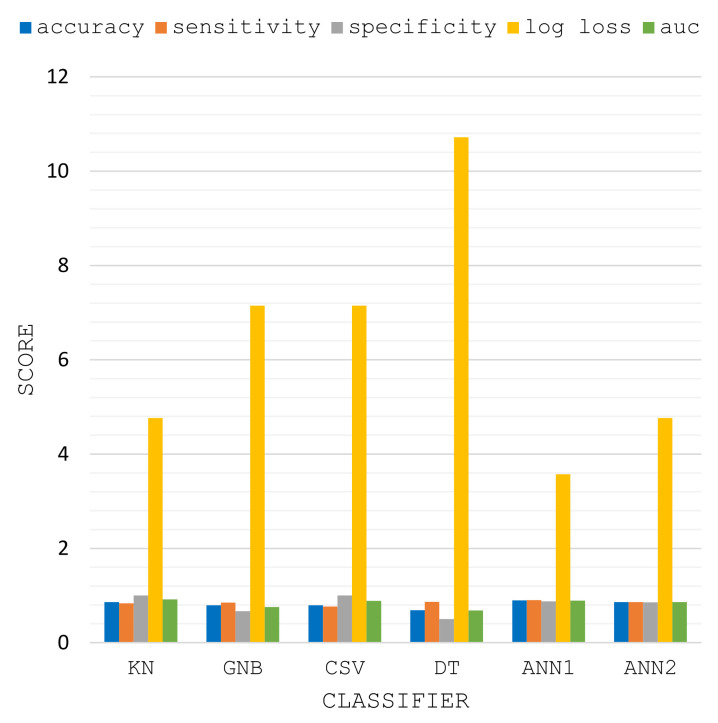
Performance scores by classifier; KN: K-Neighbors; GNB: Gaussian Naive Bayes; CSV: C-Support Vector; DT: Decision Tree; ANN1: Artificial neural network with a single hidden layer; ANN2: Artificial neural network with two hidden layers; ANN1 achieved the highest proportion of correct predictions (0.8965 accuracy), the lesser cross-entropy loss (3.573 log loss) and the highest balanced accuracy (0.8898 AUC).

**Figure 5 micromachines-14-00749-f005:**
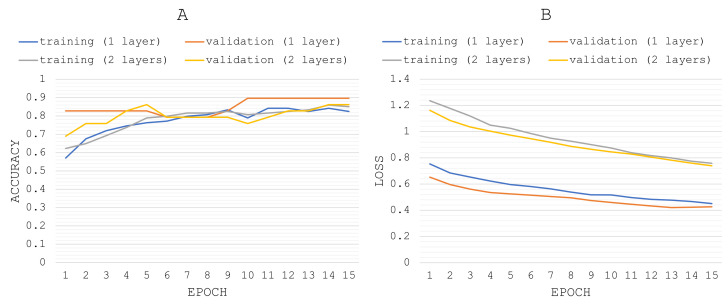
Loss and accuracy results of ANN models; (**A**): training and validation accuracy; (**B**): training and validation loss; the compilation and fitting parameters were arbitrarily configured; compilation: loss = ’binary_crossentropy’, optimizer = ’adam’ and metrics = ’accuracy’; Fitting: batch_size = 32, epochs = 15.

**Table 1 micromachines-14-00749-t001:** Features selected by RFE with five-fold cross-validation.

#	Selected Feature	Importance
1	SLC25A39	0.046594
2	YBX1	0.031416
3	NCOA4	0.028775
4	CHSY1	0.027680
5	PSAP	0.026925
6	CSDA	0.024730
7	RHOA	0.023475
8	RAC2	0.023392
9	CCDC59	0.021414
10	EIF4G1	0.021162
11	KLF13	0.020960
12	HBB	0.020262
13	HBG2	0.020044
14	NAE1	0.018367
15	LOC642210	0.018103
16	SNRPA1	0.018043
17	RPL36AL	0.017384
18	CD99	0.017245
19	YWHAB	0.017153
20	MYADM	0.015812
21	KLF2	0.015753
22	CORO1A	0.015732
23	RGS2	0.015477
24	LASP1	0.015034
25	C5AR1	0.014447
26	PCBP1	0.014407
27	HLA-DRB1	0.014128
28	NKG7	0.014030
29	RNF149	0.013935
30	ICAM3	0.013826
31	EIF3E	0.013390
32	METTL5	0.013029
33	LOC401206	0.012424
34	ITGAM	0.012246
35	LOC388720	0.011856

**Table 2 micromachines-14-00749-t002:** Efficiency and complexity results by classifier; KN: K-Neighbors; GNB: Gaussian Naive Bayes; CSV: C-Support Vector; DT: Decision Tree; ANN1: Artificial neural network with a single hidden layer; ANN2: Artificial neural network with two hidden layers; FS: Feature selection; ${}^{*}$ the differences of program size were negligible without, and with FS.

Classifier	Runtime without FS/with FS	Memory without FS/with FS	Program Size ${}^{*}$
KN	2 ms/1 ms	109 KB/55 KB	2.53 KB
GNB	2 ms/1 ms	109 KB/55 KB	2.49 KB
CSV	3 ms/2 ms	109 KB/55 KB	2.47 KB
DT	3 ms/2 ms	109 KB/55 KB	2.5 KB
ANN1	18 ms by step/14 ms by step	109 KB/55 KB	5.56 KB
ANN2	28 ms by step/19 ms by step	109 KB/55 KB	5.58 KB

**Table 3 micromachines-14-00749-t003:** Performance results by classifier; KN: K-Neighbors; GNB: Gaussian Naive Bayes; CSV: C-Support Vector; DT: Decision Tree; ANN1: Artificial neural network with a single hidden layer; ANN2: Artificial neural network with two hidden layers; FS: Feature selection.

Classifier	Accuracy without FS/with FS	Sensitivity	Specificity	Log_LOSS	auc
KN	0.6896/0.8620	0.8333	1.0	4.764	0.9166
GNB	0.7931/0.7931	0.85	0.6666	7.146	0.7583
CSV	0.7586/0.7931	0.7692	1.0	7.1461	0.8846
DT	0.6551/0.6896	0.8666	0.5	10.7189	0.6833
ANN1	0.7931/0.8965	0.9047	0.8750	3.573	0.8898
ANN2	0.7241/0.8620	0.8636	0.8571	4.764	0.8603

**Table 4 micromachines-14-00749-t004:** Descriptive statistics by classifier; KN: K-Neighbors; GNB: Gaussian Naive Bayes; CSV: C-Support Vector; DT: Decision Tree; ANN1: Artificial neural network with a single hidden layer; ANN2 Artificial neural network with two hidden layers.

	KN	GNB	CSV	DT	ANN1	ANN2
Number of samples	29	29	29	29	29	29
Mean	0.8275	0.6896	0.8965	0.5172	0.7241	0.7586
Std. Deviation	0.3777	0.4626	0.3045	0.4997	0.4469	0.4279
Std. Error of Mean	0.0713	0.0874	0.0575	0.0944	0.0844	0.0808

**Table 5 micromachines-14-00749-t005:** ANOVA test results; SS: Sum of Squares; DF: Degrees of Freedom; MS: Mean Squared; F: F ratio; *p*-value: significance level.

Source	SS	DF	MS	F(DFn, DFd)	*p*-Value
Between	2.4597	5	0.4919	2.7279	0.0213
Within	30.2972	168	0.1803	-	-
Total	32.7570	173	-	-	-

## Data Availability

The implemented pseudo codes, and the collected dataset are available at https://github.com/ponceraf2020/Pseudo-code.git (accessed on 25 February 2023).

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
