# Peer review of "A Deep Learning Approach for Predicting Multiple Sclerosis"

_micromachines, 2023, doi:10.3390/mi14040749_

Round 1

Reviewer 1 Report (Previous Reviewer 1)

The desired corrections performed and so accepted in the present form.

Author Response

Reviewer 2 Report (Previous Reviewer 3)

Reviewer 3 response to authors' response

The authors revised the interesting manuscript satisfactory other than the points below.

Point 2: In the methods section 2.1, the authors are describing GSE17048 which is whole blood mRNA expression yet there is a discussion about miRNA. I failed to see how the miRNA data was collected and how miRNA might be related to mRNA expression from this dataset GSE17048. Please clarify exactly what the samples are used in this study.

 Response 2: The expression summary values were analyzed by GEO2R, an interactive web tool that allows to view a specific gene expression through the profile graph tab. The expression values of the genes across the samples are displayed and presented as a table of genes ordered by significance, and then they are integrated into an excel spreadsheet.

Point 2 again:  Are the features used here mRNA or miRNA counts?  MicroRNA are small noncoding RNAs which have regulation properties. This is still confusing for example ‘the 74th gene (HLA-DRB15) is included in the data set’.  Is HLA-DRB15 resultant from micro RNA? See lines 90 and 91. Isn’t this a gene that is transcribed to mRNA? Again the authors need to clarify what exactly the expression values being used here are. GPL6947 is a microarray for capturing mRNA (messenger RNA).

New point 11:  In line 207, do you mean input data (5,040 data points)? The data set has only 144 samples.  So you are using 35 features with expression values across 144 samples?

New point 12:  Please correct “splitted” to “split” throughout the paper. Splitted is no longer used.  Also please check spelling throughout the paper such as in figure 1.

Author Response

Reviewer 3 Report (Previous Reviewer 4)

Some syntax errors or improper expressions exist in the manuscript.
More up-to-date studies are suggested to be cited.

Author Response

This manuscript is a resubmission of an earlier submission. The following is a list of the peer review reports and author responses from that submission.

Round 1

Reviewer 1 Report

Authors in this research used several ML classifiers for feature selection to predict multiple sclerosis.

Following are the comments based on I reject the article.

No justification to use fusion of so many classifiers to extract features/classify !

No significant contribution mentioned not achieved to publish.

Standard classifiers mixed up that cannot be considered as novelty to publish.

Overall low quality and so rejected.  

Reviewer 2 Report

I am afraid that this submission is in multiple well below the standards that I expect from a reputable journal article. The authors' approach is a rather unthinking throwing of machine learning methods - very primitive, old-fashioned and outdated at that too - at a problem, providing neither any technical contribution, nor biomedical insight. The empirical analysis is very weak, both quantitatively, with very small sample sizes not allowing for any robust conclusion to be made, and qualitatively, in that the methods used and the empirical data obtained offer no means of any novel biological insight to come from the work. Instead, we are given a bunch of off-the-shelf library routines, thrown at a small problem, with a mechanistic reporting of findings with little statistical insight or expert interpretation.

Reviewer 3 Report

This manuscript describes important research in which the authors compare four machine learning algorithms performance for classifying Multiple Sclerosis disease from non-disease samples composed of mRNA expression values collected from whole blood. Machine learning techniques can uncover not-so-obvious patterns in data sets. The paper requires extra work as it is hard to fully understand what was done to get the results.

Point 1:  Although the authors start with an omnibus dataset GSE17048 of over 30,000 whole blood transcripts, they seem to only be working with ‘29’ transcripts (I count 30-see figures 3 and 4 and related text) for the comparison of the four machine learning methods. It is not clear how they selected the 29 or 30 transcripts for the initial selection study.

Point 2: In the methods section 2.1, the authors are describing GSE17048 which is whole blood mRNA expression yet there is a discussion about miRNA.  I failed to see how the miRNA data was collected and how miRNA might be related to mRNA expression from this dataset GSE17048.  Please clarify exactly what the samples are used in this study.

Point 3: Methods section 2.2 should contain a little more detail than “a form that is most appropriate for training ML algorithms.”

Point 4:  Line 114 seems to have too many words in the sentence.    

Point 5: In line 185, it is not clear what l1 and l2 are or where they come from. Are they related to LR hyper-parameters in Table 2?

Point 6: In the figures 3 and 4, the bars seem to be unevenly spaced along the x axis. Evenly spaced bars are more pleasing to the eye and will line up nicely with the text labels.

Point 7: If I understand correctly, there were five methods used to select features (transcripts) from the ’29 or 30’ selected transcripts and then these five transcripts were used in the training step of each of the four classification techniques to classify the samples? It is not clear why only 5 transcripts were selected from the selection techniques to test each classification technique. Was there a threshhold or was it an arbitrary choice of selecting 5?

Point 8: The whole blood dataset GSE17048 contains a total of 144 samples.  In figure 5, the MLP confusion matrix indicates that only 29 samples were classified. Please describe somewhere why only 29 samples of 144 were used in the analysis. Are the samples being classified in this work from GSE17048?  If not, where did the samples come from?

Point 9: In lines 294 and 295, it is not clear what is meant by ‘choosing’ more than two configuration hyper-parameters causing the performance metrics calculation to become unstable.

Point 10: Are the references cited in the correct style for the journal? Please check the author instructions.

Reviewer 4 Report

The Abstract is suggested to be improved where the contribution and findings of the work should be highlighted.
How to initialize the agents in the proposed Algorithm?
Some additional experiments are required:
a. - Scalability
b. - Runtime
c. - Memory
d. - Sensitivity analysis

It is necessary to discuss the complexity of the proposed Algorithm.

Statistical analysis should be carried out to demonstrate that the experimental results are significant. Such as the ANOVA test and T-test

Some syntax errors or improper expressions exist in the manuscript.

More up-to-date studies are suggested to be cited.
Read and cite these references.

1.    Salamai, E.-S. M. El-kenawy and A. Ibrahim, “Dynamic Voting Classifier for Risk Identification in Supply Chain 4.0,” Computers Materials & Continua, vol. 69, no. 3, pp. 3749-3766, 2021.